# Acceptability of the FIGO Nutrition Checklist in Preconception and Early Pregnancy to Assess Nutritional Status and Prevent Excess Gestational Weight Gain: A Study of Women and Healthcare Practitioners in the UK

**DOI:** 10.3390/nu14173623

**Published:** 2022-09-01

**Authors:** Chandni Maria Jacob, Hazel M. Inskip, Wendy Lawrence, Carmel McGrath, Fionnuala M. McAuliffe, Sarah Louise Killeen, Hema Divakar, Mark Hanson

**Affiliations:** 1Institute of Developmental Sciences, School of Human Development and Health, Faculty of Medicine, University of Southampton, Southampton SO16 6YD, UK; 2NIHR Southampton Biomedical Research Centre, University Hospital Southampton NHS Foundation Trust, University of Southampton, Southampton SO16 6YD, UK; 3Medical Research Council Lifecourse Epidemiology Centre, University of Southampton, Southampton SO16 6YD, UK; 4School of Human Development and Health, Faculty of Medicine, University of Southampton, Southampton SO17 1BJ, UK; 5UCD Perinatal Research Centre, School of Medicine, University College Dublin, National Maternity Hospital, D02 YH21 Dublin, Ireland; 6Divakar’s Specialty Hospital, Bengaluru 560078, India

**Keywords:** nutrition, pregnancy, preconception, non-communicable diseases, gestational weight gain, nutrition counselling, obesity, person-centered care

## Abstract

Optimum nutrition and weight before and during pregnancy are associated with a lower risk of conditions such as pre-eclampsia and gestational diabetes. There is a lack of user-friendly tools in most clinical settings to support healthcare practitioners (HCPs) in implementing them. This study aimed to evaluate the acceptability of (1) using a nutrition checklist designed by the International Federation of Gynecology and Obstetrics (FIGO) for nutritional screening of women in the preconception and early pregnancy period and (2) routine discussion of nutrition and weight in clinical care. An online cross-sectional survey was conducted with women (aged 18–45) and HCPs (e.g., general practitioners, obstetricians, and midwives). Quantitative statistical analysis and qualitative content analysis were performed. The concept and content of the checklist were acceptable to women (*n* = 251) and HCPs (*n* = 47) (over 80% in both groups). Several barriers exist to implementation such as lack of time, training for HCPs, and the need for sensitive and non-stigmatizing communication. Routine discussion of nutrition was considered important by both groups; however, results suggest that nutrition is not regularly discussed in perinatal visits in the UK. The FIGO nutrition checklist presents a valuable resource for use in clinical practice, offering long-term and intergenerational benefits for both mother and baby.

## 1. Introduction

### 1.1. Impact of Nutrition on Pregnancy and Offspring Health across the Life Course

Overweight and obesity (measured using Body Mass Index (BMI)) can lead to complications before pregnancy, such as infertility and polycystic ovarian syndrome, and during pregnancy, such as pregnancy loss, gestational diabetes mellitus (GDM) and higher risk of intervention during delivery [1]. Conditions such as allergies, childhood obesity and congenital anomalies have also been associated with maternal obesity [1,2,3,4]. Women with a history of GDM also have a higher risk of non-communicable diseases (NCDs) such as type 2 diabetes in later life. Having a healthy, balanced diet is fundamental to body weight management, and nutrition in the preconception and early pregnancy period has profound consequences on fetal development and the mother and baby’s lifelong health [4]. Beyond weight, nutrition has a profound influence on maternal and child health. Micronutrient deficiencies (e.g., iron, vitamin B_12_, and folic acid) often exist before conception [5,6]. However, as many pregnancies are unplanned, these risks may extend into pregnancy [7]. Hence, while improved nutrition and lifestyles of women of childbearing age are generally required [8], the period before and during pregnancy offers a particular opportunity to promote this because it is a time when more women access healthcare services. If not addressed, they may worsen as nutritional requirements increase during pregnancy to meet the demands of the growing fetus. Appropriate fetal nutrition is essential for the health of the offspring such as cognitive, motor and socioemotional development, better work capacity and productivity [9,10]. Without appropriate nutrition, mothers may experience anemia or other adverse health outcomes during pregnancy and beyond.

Considering the increasing rates of obesity and overweight among women in the reproductive age group (defined for this study as 18–45 years) in both high-income and LMIC settings [11,12], preventive interventions are urgently needed. International organizations such as FIGO have recently recommended the routine discussion of nutritional needs at every contact in the clinical setting related to perinatal care [3,13]. While checklists and handouts for women that explain nutritional guidelines can improve preconception and early pregnancy consultations [14], the question is whether such a checklist or tool is acceptable to women and HCPs in the UK.

### 1.2. The FIGO Nutrition Checklist

The nutritional risk assessment checklist used in this study [15] was developed by the Pregnancy and NCDs Committee of FIGO in 2015 (Appendix A) [16,17]. The checklist was designed to be a user-friendly tool for HCPs to assess nutrition in women planning pregnancy and in early pregnancy, so that dietary and gestational weight gain recommendations could be discussed at each visit. Nutritional supplements could then be offered to women who needed them, along with a referral to diet and weight management services if appropriate.

The checklist has been tested in a pilot feasibility study in Dublin, Ireland [16,18] which consisted of 106 pregnant women and three obstetricians and gynecologists (OBGYNs). Results suggested that it helped initiate a discussion on nutrition during consultations and identified women with nutritional issues. The reliability of the checklist has also been tested in Hong Kong in comparison with validated food frequency questionnaires, suggesting that it can be effectively used to identify women with a suboptimal diet before and during early pregnancy [17]. In a recent Italian study, the FIGO checklist was used to develop a ‘periconceptional nutritional score’ as a measure of adherence to a healthy diet [19]. A higher score was associated with better pregnancy outcomes such as gestational age at birth and showed a significant association with measures of placental development. Considering the differences in healthcare systems and diet between the countries and the stakeholders involved in service delivery, a study was warranted to assess the acceptability of the checklist among UK HCPs and women, and examining the current state of practice in the UK on discussion of nutrition and weight management in the preconception and early pregnancy period.

### 1.3. Aim

The primary aims of this study were to: 1. assess the acceptability among women of reproductive age and HCPs in the UK of a nutrition checklist designed for screening and identifying nutritionally at-risk women in the preconception and early pregnancy period; 2. to understand how the checklist could be used as part of routine maternity care to stimulate routine discussion on nutrition and weight management. We also explored the current practice in antenatal care while discussing nutrition and gestational weight gain and sociodemographic differences in acceptability.

## 2. Materials and Methods

Two online cross-sectional surveys were conducted in 2021, one for women in the reproductive age group and one for HCPs. An overview of the steps is presented in Figure 1.

### 2.1. Participants and Sampling

Women living in the UK aged 18–45 were eligible. There was no limit based on pregnancy status as we aimed to include the preconception population encompassing women who were not planning a pregnancy. HCPs currently practicing in the UK who come in contact with women in the reproductive age group in routine practice (e.g., midwives, OBGYNS, registered dietitians and general practitioners (GPs)) were included. Health visitors were included as key stakeholders in the inter-pregnancy period. Participants were offered a chance to take part in a prize draw to win a voucher. As this was a pilot acceptability study to estimate the proportion of women who would find the checklist useful, a sample size calculation was not conducted.

### 2.2. Study Materials

As the original FIGO checklist was developed for a global audience, the checklist was first modified based on UK dietary guidelines and clinical recommendations for obstetrics and antenatal care suggested by the National Institute for Health and Care Excellence (NICE) (Appendix A) in consultation with UK-based registered dietitians and midwives, to ensure the new checklist would be appropriate for piloting in this study.

Though several definitions exist in the literature, for this study, the concept of acceptability as proposed by Sekhon et al. 2017 [20] was adopted as “A multi-faceted construct that reflects the extent to which people delivering or receiving a healthcare intervention consider it to be appropriate, based on anticipated or experienced cognitive and emotional responses to the intervention.” Relevant factors assessing acceptability (affective attitude, burden, self-efficacy, perceived effectiveness, and intervention coherence) were included based on the theoretical framework [20] used by previous pilot studies [16].

Two separate questionnaires were developed (Appendix A) modifying the tools previously used to evaluate the acceptability [16]. Both questionnaires were specifically designed for each target group (women, HCPS) and contained key sections for: 1. sociodemographic details; 2. views on the FIGO Nutritional Checklist (e.g., The checklist will be easy to complete in a waiting room of a GP clinic/midwife’s clinic) and on the importance of nutrition in the periconception period (e.g., Changes to my diet before I become pregnant can help make sure my child is born healthy); 3. experience during clinical visits and perceptions about discussion of nutrition routinely (e.g., The HCP had enough time to talk about my diet with me). The questions in each section were a combination of open-ended questions and a mixture of 5-point Likert scale questions (ranging from strongly agree to strongly disagree) with option-based questions for each target group. Sociodemographic data were also collected as dietary behaviors, health literacy and obesity are associated with ethnicity, educational status and socioeconomic status [6,21].

### 2.3. Data Collection

Ethics approval was obtained prior to data collection from ERGO University of Southampton (ERGO 61724.A2.R4, ERGO: 61736) and Health Research Authority for HCPs (NHS REC reference: 21/HRA/3972). For both surveys, a mixture of sampling techniques was used to obtain a wider representative sample across the UK. Convenience sampling was used by disseminating the survey in the Southampton General Hospital. A snowball strategy was also used where participants were encouraged to forward emails to relevant groups and through retweets. Purposive sampling was conducted using social media (Twitter, posts on relevant Facebook groups and advertisements on Facebook), professional networks, charitable organizations (e.g., Diabetes UK, British Dietetic Association) and websites such as Mumsnet. The survey was conducted on Microsoft forms (Microsoft Corporation, Redmond, WA, USA) (for HCPs) and Survey Sparrow (SurveySparrow Inc., Palo Alto, CA, USA) (for women).

### 2.4. Analysis

#### 2.4.1. Quantitative Analysis

As the survey participant numbers were low for the HCPs, only descriptive statistics are presented for this group. For the women’s survey, the quantitative data gathered from the online platform were entered into the statistical package SPSS (IBM Statistics V.28, IBM Corp., Armonk, NY, USA). Incomplete responses were included only if the respondent had completed at least one section from the questionnaire in full along with information on pregnancy status and other key demographic variables. Categorical variables were created for analysis. For example, postcode data were converted to Index of Multiple Deprivation (IMD) ranks and deciles [22] and later recoded into quintiles to explore associations between demographic characteristics and acceptability. IMD is the official measure of relative deprivation for small areas developed by the Ministry for Housing, Communities and Local Government (MHCLG) in England. The national deprivation deciles rank all Lower Super Output Areas (LSOAs) in England from the most deprived (rank 1) to least deprived (rank 32,844) and then divide them into ten categories (deciles) across the whole of England). For the final variable, IMD quintiles were derived with quintiles 1 = most deprived and 5 = least deprived.

For our study, we adopted the definition of preconception using a life course approach [4,23], considering the high rates of unplanned pregnancies globally, and as women who are not active planners may have future pregnancies. Responses to the women’s survey were divided into three categories of preconception, pregnant and up to 2 years postpartum. This was to obtain an overview of the different HCPs whom the women had met for nutrition and pregnancy-related reasons and to explore any differences in acceptability. Though the preconception subgroup could include women who were more than two years postpartum, the data on past pregnancies were not collected as we aimed to evaluate the most recent practices in the healthcare system. Throughout the results section, ‘*N*’ denotes the total valid number of participants included in the analysis and ‘*n*’ depicts the frequencies for the variables being described. Valid percentages are presented in the findings for questions with missing data.

A simple binary logistic regression was conducted to explore the relationship between all independent variables and two outcome variables developed for the following questions on the checklist: The checklist will be easy to complete in a waiting room—Yes/No; Overall, the information on both pages of the checklist will improve discussions of nutrition with a healthcare professional—Yes/No. These two questions were selected as they summarized the overall acceptability of the checklist. Though various combinations of independent and outcome variables were assessed by regression analysis and Chi-square tests, the results were not informative or had extremely wide confidence intervals due to the small sample size. Hence, this paper only presents the descriptive results in detail and key findings of the regression analyses are reported where differences were apparent.

Data from the overall Likert scale were treated as ordinal. Research has shown that a “neutral” response could mean indifference to the topic of research under study. It may also be selected by participants who feel it to be less confrontational than “disagree” (social desirability), or who are hesitant [24,25]. Hence, when transformed into binary variables for the regression analysis and in the discussion below, neutral responses for this study were included in the “disagree and strongly disagree” groups as it was considered inappropriate to assume that respondents who answer “neutral” would accept the checklist or other questions related to nutrition. This dichotomization was only performed to check for associations between background characteristics and the acceptability of the checklist.

#### 2.4.2. Qualitative Content Analysis

Three free text boxes were presented in with the following questions—“Please add any comments and suggestions to improve the checklist (for HCPs and reproductive aged women), “How would you feel if nutrition diet and weight gain during pregnancy were discussed routinely in all antenatal appointments, even if it was not the main reason for your visit? (women only)” and “What are the issues (if any) that you face while supporting women with dietary behavior change or prevention of obesity? (HCPs only)”. Responses to the data warranted separate analyses for each group, so separate coding frameworks were developed for each question and survey. The content analysis was conducted using Robson’s Six Step framework in the three phases of preparation, organization and reporting as discussed by Elo and Kyngäs [26,27]. The inductive content analysis was broadly performed by open coding, and categorization followed by abstraction. The codes were later combined according to meaning into sub-categories and finally into categories. To test the clarity of any category/code definitions, and enhance the trustworthiness of the results, the codes were tested on samples of text by a second reviewer (CMcG) and refined after discussion [28]. The research was underpinned by a critical realist [29] philosophical approach which allowed to capture the meaning in the data. The frequencies are not presented for the number of respondents, which is usually conducted in a positivist position. The number of coded comments is presented to provide a picture of the distribution of sub-categories without assigning a particular weight to a certain category.

## 3. Results

### 3.1. Participant Characteristics

Overall, 251 responses were included for the women’s survey and 47 for the HCP survey. Data for IMD were available for 147 women who provided their postcode.

For the women’s survey, the mean age of participants was 32 years (range 19–45). Participants’ characteristics are shown in Table 1. Finally, many pregnant women (42%) answering the survey were in the second trimester, and hence would have completed their first appointment with the midwife or GP. This was followed by women in the third trimester (32%) and women in the first trimester (26%). Most preconception women were from the younger age groups.

Twenty-four women (9.6%) reported having medical conditions/co-morbidities during pregnancy or high-risk pregnancies. The most common conditions included hyperemesis gravidarum (*n* = 4), gestational diabetes (4), high blood pressure (*n* = 4), and type 2 diabetes (2). Thirty women (12%) reported having special dietary requirements, for example diabetic diet (*n* = 6), food allergies (4), gluten intolerance (4), vegan/vegetarian (*n* = 2), and eating disorders (*n* = 2). Vegan and vegetarian diets may, however, be underreported as the question was about “dietary requirements not addressed in the checklist”.

A total of 47 HCPs completed the second survey (summarized in Table 2). Respondents predominantly consisted of midwives (43%) followed by OBGYNs (17%) and dietitians (17%).

### 3.2. Acceptability of the Checklist

Results from the survey of reproductive-aged women showed that most respondents would recommend the checklist for both preconception and pregnancy (Figure 2). A large majority of women also agreed that the format, style and content were easy to understand, and that the checklist would be easy to complete in the GP waiting room or midwife clinic.

A large majority of women within each reproductive group considered the checklist to be easy to complete (78% for preconception; 82% for pregnant; 94% for postpartum groups). Overall, only 32% of women reported that they had not thought about the questions in the checklist before pregnancy, suggesting a high awareness of nutrition-related topics among the respondents. However, on subgroup analysis, it was seen that 47% of preconception women agreed that they had not thought about the questions in the checklist, followed by postpartum (24%) and pregnant (19%) women, indicating that awareness about nutrition before and during pregnancy was lower in the preconception group. Women with post-graduate or university degrees and from the 4th and 5th IMD quintiles were more likely to have considered diet for pregnancy, compared with non-college-educated and respondents from more deprived areas—however, the numbers in the subgroups were too small for meaningful conclusions to be drawn. Almost 52% of all women said they felt better prepared to adopt a healthier lifestyle during or before pregnancy after reading the checklist. Overall, 70% of women agreed that information on both pages of the checklist would help in visits with HCPs, indicating high acceptability for use of the checklist.

Figure 3 illustrates the acceptability of the checklist for HCPs in the UK. Like the results from the women’s survey, the majority of HCPs felt that the content and format of the checklist were easy to understand and use; however, only 13% of HCPs agreed they would be able to cover the contents during routine appointments. Most HCPs (89%) agreed that it could help initiate a discussion on nutrition, that it covered all important nutrition-related topics (85%) and that it could be used for routine health promotion, even when women visit for other issues, if they complete it in advance (66%).

On cross-tabulation by job titles, most OBGYNs (75%) agreed it would be easy to use, compared with approximately half of the dietitians, community and staff midwives. All midwives and GPs, 75% of OBGYNs and 88% of dietitians reported that they would not have enough time to discuss the checklist in their clinics. More than 60% of all staff groups (except dietitians) said they would recommend the checklist for routine nutrition health promotion. However, most dietitians disagreed on the use for routine health promotion, potentially because dietitians were more aware of wider nutritional issues not included in the checklist or the need for modification of content in the checklist. When compared with the open-ended responses, they were also more likely to critique the checklist as being too focused on weight/BMI.

#### 3.2.1. Findings from Regression Analysis

Compared with the youngest group, women aged 31–35 had twice the odds of agreeing that the checklist was easy to complete (odds ratio (OR) 2.87 (95% confidence interval (CI) 1.05–7.83; *p* = 0.039)). Postpartum women had more than four times the odds (OR 4.7 (95% CI 1.35–16.49, *p* = 0.015)) of agreeing that the checklist would be easy to complete in the waiting room compared with preconception women. Finally, compared with women from the most deprived regions, women from quintiles 4 (OR 4.7 (95% CI 0.9–24.36)) and 5 (OR 3.8 (95% CI 0.6–17.1)) had higher odds of agreeing that the checklist was beneficial, although the confidence intervals were very wide (*n* = 147).

#### 3.2.2. Suggestions for Modification and Implementation of the Checklist

Seventy-seven responses to the open text question from the women’s survey informed the development of five key categories. Appendix A presents a summary of the key categories developed from the women’s responses. Women recommended that certain topics needed to be included in further iterations of the checklist, such as vegan/vegetarian diets, intake of caffeine and examples of portion sizes. In addition, the issues of exercise during pregnancy and the impact of hyperemesis on diet were highlighted.

Several participants added that topics in the checklist should be discussed without blaming the mother or causing stress.

“During my first pregnancy, the midwife couldn’t advise on whether I could continue the exercise regime I was doing because she didn’t know… you are advised of everything you can’t do … but not what you can actually do”(ID 1050)

Providing adequate information for women to review independently post-consultation was also recommended.

“I believe more supportive tools to take away from the meeting along with the communications provided at the meetings.”(ID 1044)

HCPs provided information related to two key categories (34 responses). The first category captured discussions related to issues hindering the operationalization of the checklist in routine practice in the UK such as a lack of time, women feeling judged, and a lack of confidence in discussing the contents of page 2 of the checklist, e.g., polyunsaturated fatty acids (PUFAs). Referring to lack of time as a barrier (discussed further) HCPs also suggested that -

“It might be better to concentrate on one aspect of the checklist that is a priority for the patient, would also be more patient-centred. e.g., physical activity—could talk about options…”(GP, 101)

The second category represented revisions suggested by HCPs in order to make the checklist more user-friendly for patients and HCPs, as well as modifications needed to align the checklist with UK NICE guidelines and to add further relevant information. The suggested changes included the addition of translated versions of the checklist to a range of languages, simplification of contents to imperial units, as well as detailed revisions to make it more gender inclusive, appropriate for people with low literacy levels and revising the content related to weight gain to be less confrontational or judgmental.

### 3.3. Response to Questions on Current Practice

Approximately two-thirds of women (67%) replied that they had not completed a nutrition checklist or a dietary assessment tool resembling the FIGO checklist during a consultation (67%; *N* = 216), but 12% of respondents had never visited an HCP for fertility/pregnancy-related reasons. Women who had completed a similar checklist recalled using it at midwife consultation (*n* = 13); with their GP (*n* = 3); a nutritionist (*n* = 1); at fertility assessment clinics (*n* = 2); or on an independent online website (*n* = 1). Very few women (less than 36% from each reproductive group) recalled discussing diet or nutrition during their visits to the HCPs indicating a missed opportunity in routine care. Women who had such discussions had discussed this mostly with their midwives (*n* = 42), GPs (*n* = 10), OBGYNs (*n* = 5) or nurses (*n* = 4). Similarly, only 26% of women reported that their HCPs discussed weight/weight gain during pregnancy with them (*N* = 223), usually with midwives, but some discussed them with their GPs.

Figure 4 illustrates the agreement on the questions presented in the survey about women’s experiences during routine visits.

Interestingly, less than 32% of women within each reproductive group perceived that discussing nutrition/weight with HCPs was easy, suggesting a high level of uneasiness among women about discussing these topics in healthcare settings.

A key finding from the survey, indicating the gap in similar tools in current practice, was that 92% of HCPs stated they were not using a similar checklist or tool for nutrition in pregnancy in their routine practice. The tools mentioned (*n* = 4) included the ones from their antenatal booklets and leaflets from the Food Standards Agency. While a vast majority of HCPs (95%) agreed that discussing nutrition is important during pregnancy, half the respondents found it difficult to initiate discussions related to weight (50%) and nutrition (43%) in clinical practice. HCPs also felt more confident discussing nutrition (68%) compared with weight management (51%), with only 38% reporting they had the necessary tools/training needed for discussing both these topics with women routinely (Figure 5).

While discussing their current practice, only 26% of HCPs reported that women planning a pregnancy may initiate conversations related to nutrition in consultations. Less than half (43%) of HCPs reported initiating the conversations on nutrition/weight regularly, indicating that nutrition may not be discussed at all in most of the consultations in maternity care. While more midwives reported that discussions on nutrition and weight are difficult to initiate, dietitians reported the most confidence in nutrition (100%) and weight management (62%) and access to tools compared with other groups. More than half the HCPs (57%) reported that they do not meet women in the preconception period in the clinical setting, when asked if women approach them for pregnancy planning.

### 3.4. Discussing Nutrition, Weight, and Obesity-Related Topics in Clinical Care Routinely

From the 241 valid responses coded from the survey for women in the reproductive age group, most suggested that a discussion on nutrition and healthy weight gain during each appointment would be beneficial overall. As some women may hesitate to initiate the discussion themselves, the questions raised by the HCP would help them request clarification. Women also described how they valued personalized care and how raising the topic during different periods of pregnancy could help with specific issues, such as hyperemesis, or high-risk conditions, e.g., gestational diabetes. Overall, HCPs were perceived as reliable and trusted sources of information. It was reported that, while there are many sources online, very few are reliable and they may contain incorrect information or not have information at all on some issues such as healthy weight gain and sources of nutrients. However, some participants also felt that routine discussion could induce stress and seem intrusive or judgmental, suggesting that how these issues were raised and communicated was important. The booking appointment (first appointment with the midwife in the UK) was suggested as an appropriate time for a conversation on nutrition, as other appointments are often shorter or may include topics such as mode of delivery which were considered to be more important during later pregnancy. Table 3 below presents a summary of categories and sub-categories with illustrative quotes from the women’s survey.

In contrast, fewer HCPs supported the routine discussion of nutrition. The categories that arose from 34 valid responses from HCPs outlined common barriers faced while discussing nutrition/weight/obesity-related topics with women in maternity care and these are presented in Table 4. HCPs felt that they are often required to focus on acute emergencies or high-risk pregnancies that present to the clinic. Nutrition may not be a high-priority topic to be raised in routine appointments in the perinatal period. Similar to the quantitative results, several HCPs raised lack of time as a factor that affects how often they discuss nutrition, with short consultation time affecting which topics they discuss. HCPs also perceived the lack of access to women in the preconception/postpartum groups as a barrier.

It was suggested that the information on the checklist should consider cultural sensitivities, and that there may be difficulties in having a conversation about nutrition with patients who may not be fluent in English. Similarly, weight and BMI were key components of the checklist that needed to be communicated appropriately on a case-by-case basis, considering the patient’s lifestyle and other risks to avoid negative implications. Some HCPs also did not feel confident about delivering the support required for weight loss, or management of special diets and communicating this sensitively with patients. Requests for resources and training were also raised. There was a clear preference by HCPs for further digital resources or an online version of the checklist with information that women could access.

## 4. Discussion

The current study was the first in the UK to test the acceptability of the FIGO nutrition checklist with women of reproductive age and HCPs. For the primary aim, findings suggest that the concept of the checklist was acceptable to women and HCPs, and more than half of them felt better prepared to adopt a healthier lifestyle during or before pregnancy after reading the checklist. Results of the secondary aims suggest that discussions about nutrition and weight management were potentially missed in most consultations and while routine discussion of these topics were considered helpful, they need to be conducted in a sensitive manner. This study also advances the understanding of barriers in the healthcare system which could affect implementation, and provides recommendations informed by clinical stakeholders and women in the reproductive age group. Finally, our study also strengthens the evidence for inequalities in health literacy and knowledge among women in the reproductive age groups shown in previous research [30], as women of higher educational status (college and above) and from less deprived areas were more likely to think about their diet before pregnancy. Thus, the implementation of such tools needs to be considered in the context of the wider social determinants of health, and factors beyond clinical health systems.

### 4.1. Recommendations for Implementation of the FIGO Nutrition Checklist

As the checklist was considered acceptable and important for initiating discussions, and studies evaluating its validity and feasibility have shown promising results, it is recommended that the modified UK-specific checklist be completed by women either online or on paper in a clinic/hospital. To address the barrier of time, HCPs recommended using online resources, and the provision of the checklist in advance of the women attending clinics. Digital versions of the checklist and mobile applications were suggested by the HCPs and some women, and these have shown promise in the preconception and pregnancy period for issues such as smoking cessation and lifestyle modification [31,32]. In addition, women also expressed that due to unreliable and untrustworthy information online, they would prefer to have information from HCPs. As health literacy can influence behaviors in relation to seeking information online [33], digital interventions could potentially increase health inequalities if they are not easily accessible and supported by targeted dissemination to women from disadvantaged backgrounds or with low digital literacy skills [31,33]. They would also need to be available in multiple languages. While the aim of the FIGO checklist was to develop a minimum nutritional adequacy screening tool for maternal and fetal health, some of the suggestions from participants indicate that they would prefer inclusion of wider issues such as eating disorders, caffeine intake and portion sizes. Further iterations of the checklist could include these issues and have potential adaptations for high-risk pregnancies such as GDM. The checklist could include consideration of special diets and be available in different languages. Recently FIGO has taken forward efforts to develop a web-based version of the checklist which can be operationalized in multiple countries considering some of the factors discussed above [34].

Figure 6 presents a model for integrating the checklist and associated discussions into routine care, considering the factors discussed in this study. This could be merged with the recent agenda for improved preconception health encouraging routine screening for pregnancy intention [7,35], to address nutritional needs before pregnancy such as folic acid intake and achieving healthy weight before conception. The responses could then be discussed during consultation. However, to make the consultations effective and suitable, Figure 6 also summarizes key points for HCPs and promotes patient (person)-centered care. Training HCPs in Healthy Conversation Skills [36] and in diet and nutrition guidelines would complement the delivery of information through the checklist, as well as facilitate productive, non-confrontational conversations post-consultation. Incorporating the importance of NCD prevention and nutrition in the educational curriculum for medical and allied health professions is also imperative.

### 4.2. Recommendations to Support Discussion on Diet and Nutrition in the Perinatal Period in the UK

Multiple factors emerged from the findings, stated by both women and HCPs, which could either deter or encourage discussions on nutrition or diet and how these would impact on the patient. Interestingly, few women recalled discussing diet/nutrition and gestational weight gain during their visits to HCPs, and some reported that HCPs were not able to provide adequate information, especially for exercise-related guidelines. This could be influenced by the problems raised by HCPs about initiating discussions on weight management and diet, concerns about inducing anxiety in patients, and the call for more resources and training. Such reservations about nutrition/weight discussion have been expressed in wider research by UK-based HCPs (midwives) who also reported difficulty in initiating and discussing obesity in current practice [37]. The qualitative study found that while developing a strong midwife–woman relationship was a high priority, most midwives stated that patients responded negatively when discussing obesity. Overall, HCPs wanted better training to initiate discussions, in using sensitive language and communicating risk in order to empower women. This calls for better support through continued professional development and other activities for a broad range of stakeholders who meet women in the reproductive age group [38].

Poor communication skills of the HCPs could also act as a potential barrier to discussing weight and nutrition routinely. A possible way to address this issue is through patient-centered care, that the HCPs also endorsed. This might involve, among other aspects, a focus on what women can do over what they should not do [39]. Experts have recommended that patient-centered care in pregnancy should move to person-centered care [40] which considers the patient’s values and beliefs and includes three key components: 1. Initiating the partnership (considering the individual’s wants, goals and motivation and impact on their life); 2. Developing and working on the partnership to achieve the agreed goal; 3. Safeguarding the partnership through documentation (detailing the individual’s care preferences in medical records) [41]. An evidence-based technique to support dietary and lifestyle behavior change in pregnancy is Healthy Conversation Skills, which uses an empowerment approach to support goal-setting, behavior change and to improve interpersonal communication [36]. Evaluation of the program, which stresses listening over the provision of information, has shown high acceptability among women and HCPs in the UK [36].

Women’s confidence in their own knowledge and health behaviors discovered in this study seemed to be a factor leading to non-acceptance of the checklist and regular discussion of nutrition. In previous research with pregnant women, Morris et al. (2020) [42] defined “health identity” as the extent to which women perceived themselves as healthy. This could range from ‘Health focused’ where women feel confident in their knowledge and health status to ‘Health disengaged’ where they do not consider their health to be a high priority. Both these groups may be harder to engage, as women could be less likely to be interested in lifestyle support, while women in between these extremes were more likely to be interested in behavior change. Studies have shown that health identity could influence the uptake of health messages [43]. Using Healthy Conversation Skills such as open discovery questions to explore women’s lives and beliefs could potentially increase engagement in these topics as well as providing non-judgmental care. Our findings are broadly similar to studies where women with obesity before and during pregnancy have reported facing weight bias or stigma during appointments with GPs, midwives and other HCPs, with severe implications for the woman’s health and wellbeing [38]. Research has shown that weight stigma can have differential effects based on gender, with women more likely to experience it, and to report lower levels of motivation and physical activity as a result [44], along with increased discomfort and potential reinforcement of self-loathing and body image issues [38]. Using patient-centered care, HCPs can recognize and respect what the patients wish to discuss.

Our findings also highlight the need to disseminate messages for preconception nutrition to the right target group as many of the women in the survey reported planning a pregnancy in the next year. On the other hand, barriers such as unplanned pregnancies and HCP perceptions that they did not have the opportunity to deliver preconception care were also evident. While public health messaging and health promotion also have a role to play in the nutrition and wellbeing of women who may not approach HCPs, awareness-raising among HCPs is also needed in order for them to address nutritional risks before pregnancy and consider it a priority in antenatal visits. Finally, considering the myriad of barriers, and potentially specialist services required for management of NCDs, it is not fair to assume that the burden of preventive public health for NCDs should be addressed by clinicians or by individuals alone. For example, issues related to health inequalities and low health literacy levels will have to be tackled with support from public health agencies, educational settings and the government. Future research should also consider evaluating the cost-effectiveness of the checklist and its effectiveness on lifestyle modification, pregnancy outcomes and dietary behavior change.

### 4.3. Strengths and Limitations of This Study

This study provides several practical recommendations for inclusion in clinical practice informed by patient and HCP lived experiences. We adopted a user-friendly design for the survey to improve completion rates and data were checked and cleaned to include only incomplete survey responses which provided adequate meaningful data. In addition, diverse modes of participant recruitment including wide dissemination of Facebook advertisements across the UK were adopted to reduce selection bias. While all efforts were made to design and disseminate the survey appropriately, some limitations exist.

The main limitation has been the small sample size for the quantitative analysis. Studies have shown a lower response rate to surveys both online and in-person during the period of the pandemic [45]. Considering the context of the pandemic and lockdown during the period of the survey, non-response could have potentially occurred due to restrictions in data collection in person and a rise in research studies using online survey-based methods leading to survey fatigue [46]. The rate of completion for our survey was high ((96%) for women who started it (measured automatically by Survey sparrow)). This suggested that the survey content and questions were well-designed and of an appropriate length but that the response rates in both groups were more likely affected by survey fatigue during the pandemic. Another limitation is that most women completing the survey were university educated, of predominantly white ethnicity and from less deprived regions (when data were available). While efforts were made to disseminate the survey to a range of community groups on social media, our findings may suggest that women of higher-education and socioeconomic status were more likely to access social media websites. The findings of the inferential statistics are presented in the context of these limitations and the low sample size which limits the validity of the quantitative results.

## 5. Conclusions

In conclusion, results of a UK-wide survey show that HCPs and women in the reproductive age group found the FIGO nutrition checklist to be acceptable for use for clinical visits before and during pregnancy. This study also provides further recommendations to support implementation. The current state of practice in antenatal care and women’s experiences while discussing nutrition-related topics before and during pregnancy were explored using mixed methods, showing a gap in existing practice for nutritional screening in antenatal care. This needs to be urgently addressed considering rising rates of maternal obesity and GDM. A range of HCPs such as OBGYNs, midwives and GPs can adopt the checklist to support women in the perinatal period to improve their nutritional status using a person-centered approach to discuss nutrition and weight management.

## Figures and Tables

**Figure 1 nutrients-14-03623-f001:**
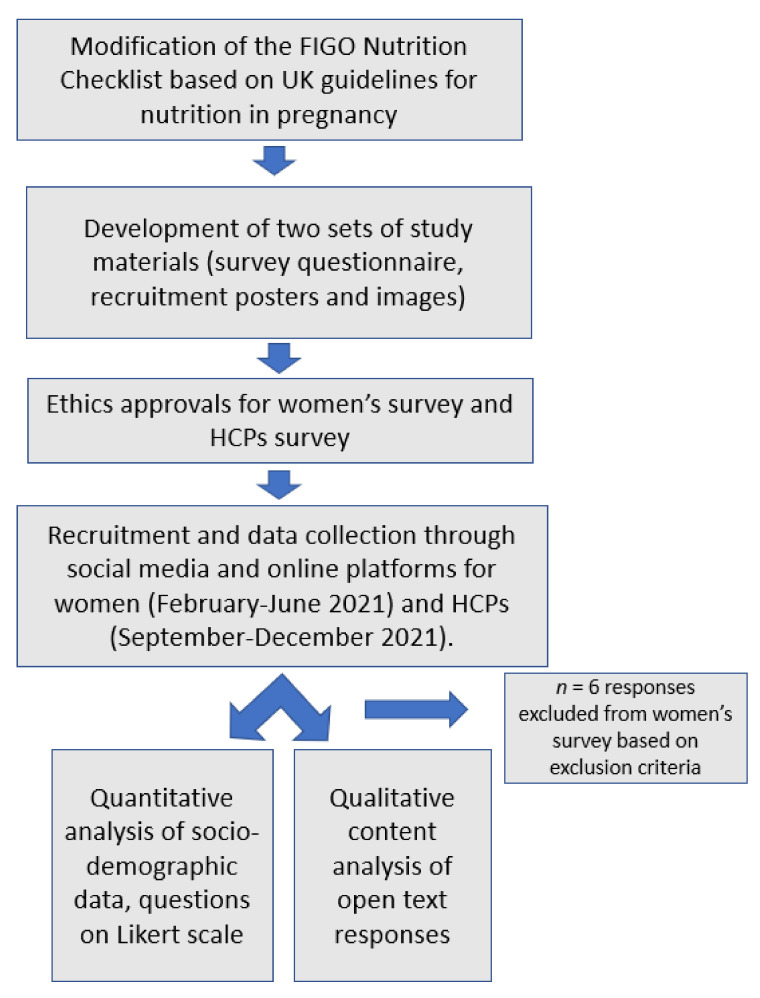
Flowchart for study process. FIGO, International Federation of Gynecology and Obstetrics; HCPs, healthcare practitioners.

**Figure 2 nutrients-14-03623-f002:**
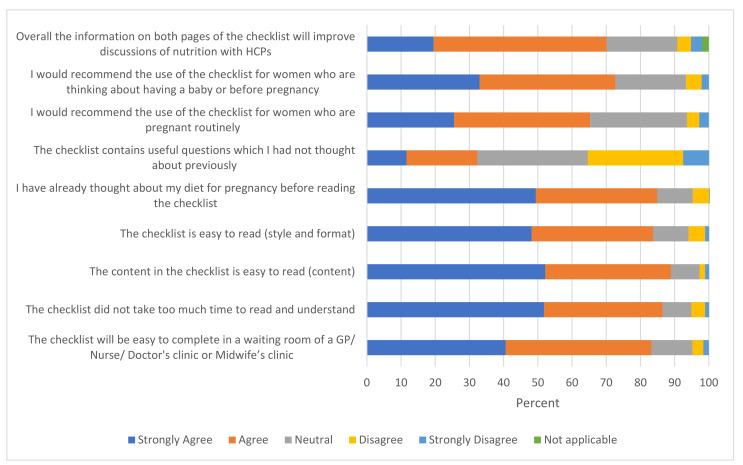
Findings for the questions on acceptability of the checklist from a survey of women in the reproductive age group (total sample *N* = 251; for the question on “recommends the checklist before pregnancy” *N* = 194; “thought about diet before pregnancy” *N* = 231). GP, general practitioners.

**Figure 3 nutrients-14-03623-f003:**
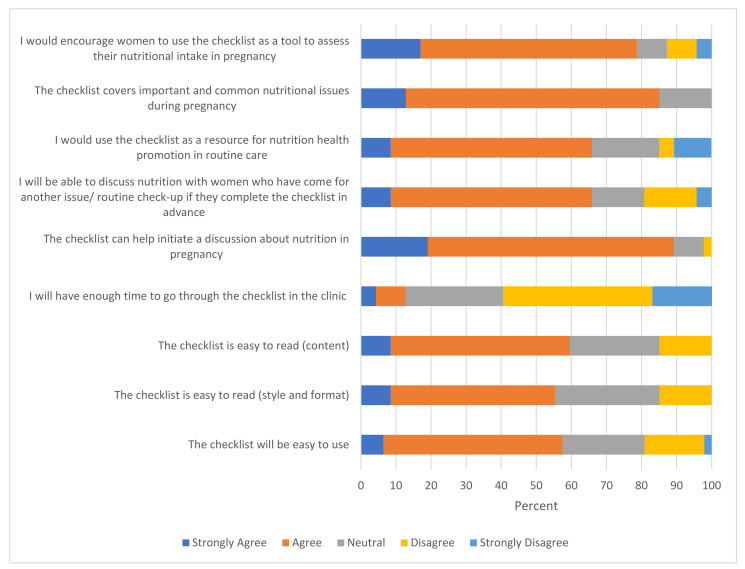
Findings for the questions on acceptability of the checklist (HCPs survey).

**Figure 4 nutrients-14-03623-f004:**
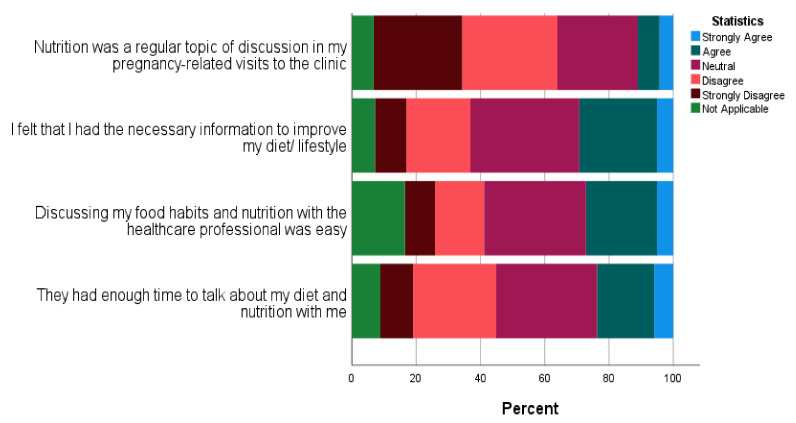
Stacked bar chart on experience of discussing nutrition and weight with HCPs based on the survey of women in the UK (*N* = 236) (not applicable indicates that women responded that they were not pregnant/did not have a baby recently/were not thinking about a pregnancy).

**Figure 5 nutrients-14-03623-f005:**
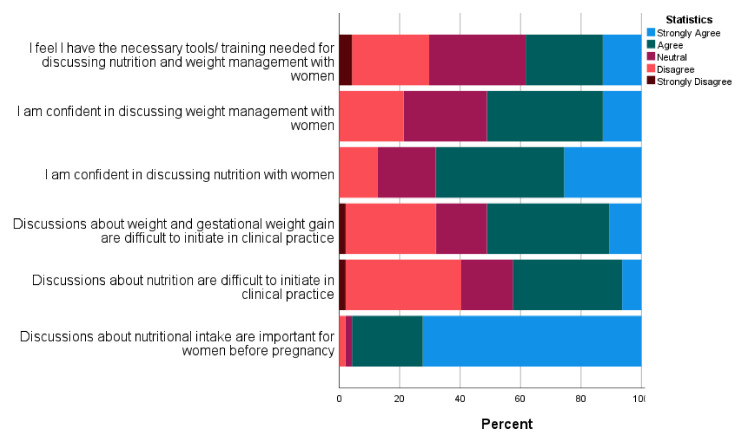
HCP perceptions on discussion of nutrition in routine periconceptional care.

**Figure 6 nutrients-14-03623-f006:**
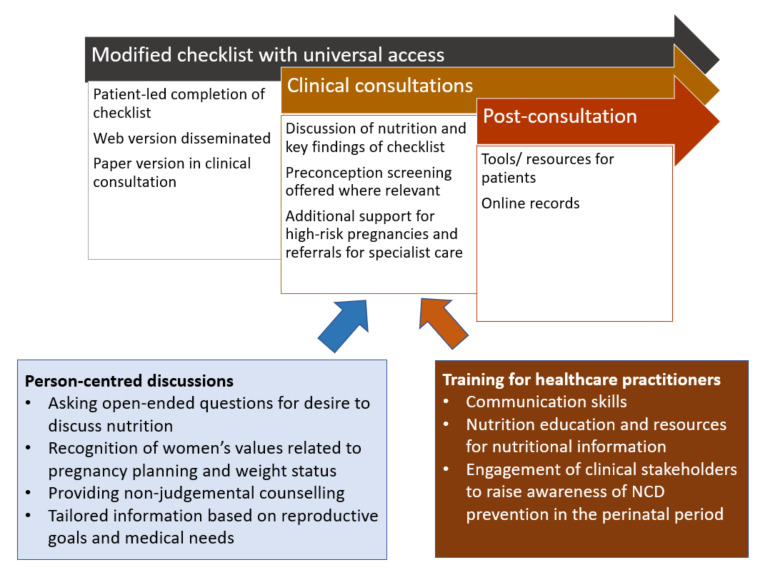
Recommendations for implementing the modified checklist during clinical visits. NCD, non-communicable diseases.

**Table 1 nutrients-14-03623-t001:** Demographic characteristics of survey participants (women).

Descriptor	*N*	%
**Reproductive Group**		
Preconception	106	42.2
Pregnant	91	36.3
Postpartum	54	21.5
**Age, years**		
18–25	35	13.9
26–30	61	24.3
31–35	93	37.1
36–40	36	14.3
41 and above	26	10.4
**Region**		
East of England	24	9.6
East Midlands	9	3.6
London	25	10
Northeast	9	3.6
Northwest	19	7.6
Southeast	64	25.5
Southwest	32	12.7
West Midlands	17	6.8
Yorkshire/Humberside	18	7.2
Northern Ireland	11	4.4
Scotland	17	6.8
Wales	6	2.4
**Education**		
Primary school or Secondary school up to 16 years	6	2.4
Higher or secondary or further education (A-levels, BTEC, etc.)	26	10.4
College or university	117	46.6
Post-graduate degree	89	35.5
Other	4	1.6
Prefer not to say	9	3.6
**Ethnicity**		
White	212	84.5
Black/Black British	6	2.4
Asian/Asian British (South Asian)	18	7.2
Asian/Asian British (Chinese)	4	1.6
Mixed/Multiple ethnic groups or Other	9	3.6
Prefer not to say	2	0.8
Index of Multiple Deprivation Quintile *	(*N* = 147)	(Valid percent)
Quintile 1	15	10.2
Quintile 2	28	19.0
Quintile 3	32	21.8
Quintile 4	30	20.4
Quintile 5	42	28.6

Total—251; * quintile 1 = most deprived and quintile 5 = least deprived. BTEC, Business and Technology Education Council.

**Table 2 nutrients-14-03623-t002:** Key characteristics of HCPs.

Descriptor	*N*	%
**Staff category**		
General Practitioner	3	6
OBGYN	8	17
Staff Midwife	11	23
Community Midwife	9	19
Health Visitor	5	11
Dietitian	8	17
Other	3	6
**Years of Clinical experience ***		
Currently training/less than 2 years	6	13
2–5 years	8	17
6–10 years	6	13
More than 10 years	27	57
**Region ***		
London	6	13
North (East and West)	9	19
South (East and West)	23	49
East of England	1	2
Scotland	3	6
Wales	4	9
Northern Ireland	1	2

* groups in some categories have been merged due to low numbers. OBGYN, obstetricians and gynecologists; HCPs, healthcare practitioners.

**Table 3 nutrients-14-03623-t003:** Summary of content analysis of women’s views on routine discussion of nutrition.

Category	Sub-Categories (*n*)	Sample Quotes
**A. Positive views on routine discussion**	Beneficial (127) Helpful before and after pregnancy (10) Personalised care (21)	“When pregnant you have no idea if you’re eating the right things/gaining the right amount of weight and I found I received next to no advice or had any conversations around this.” (ID 1022)
**B. Negative implications of routine discussion**	Inappropriate for routine visits (26) Induces stress (7)	“It doesn’t need to be discussed routinely as it would cause unnecessary stress … Pregnancy sickness and nausea affect dietary habits in the beginning and keeping any food down is a success so regular discussions about nutrition would cause additional pressure.” (ID 1072)
**C. Information**	Lack of Information on good nutrition (16) Reliable source needed (4)	“I’d find it really helpful. I was vegetarian and chose to eat fish pre-pregnancy to help improve my diet. Any discussion around diet would have been immensely helpful and reassuring.” (ID 1113)
**D. Communication and conversation**	Sensitively framed conversation without judgement (21) Excessive focus on weight/BMI (7) Time as a barrier (10)	“I would be happy with this as long as it was framed positively, not in a way that makes mums feel they are damaging their baby or being shamed. E.g., statements like eating fatty acids are good for baby’s brain development” (ID 1027)
**E. Other (19)** (This separate category looks at reasons to refuse routine discussion for oneself but accept it for other women or for the health of the baby.)	Confidence in own knowledge and health behaviors (13) Nutrition education not seen as the main priority during pregnancy (6)	“I didn’t gain much weight during/after my pregnancy and already back to my pre pregnancy healthy weight. Nutrition was never mentioned to me, but I already live a healthy lifestyle. I think it would be very beneficial to others though.” (ID 1026)

*n* = number of coded comments. BMI, body mass index.

**Table 4 nutrients-14-03623-t004:** Summary of results content analysis on HCP’s views on routine discussion of nutrition.

Category	Sub-Categories (*n*)	Sample Quotes
**A. Practical issues relating to discussion of nutrition**	Requirement to focus on higher priority health-related topics in perinatal care (3) Time for appointments (15) Barriers in the healthcare system (10) Access to patients (3)	“Often other priority areas e.g., communicating info about gestational diabetes and hypertension. As always obesity gets underprioritized” (GP 101)
**B. Concerns around communication relating to nutrition and the checklist**	Checklist information should be tailored to patient needs (5) Topics discussed may have negative implications (10) HCPs not confident to discuss areas of the checklist or delivering support required (6)	“… By changing the focus of the dietary changes from being all about weight, to being about making sure mother is having the right nutrition for her and baby, then we can still achieve the same healthy dietary changes (and therefore outcomes) but without the negative connotations and stigma of discussing weight all the time.” (Dietitian, 144)
**C. Support for routine discussion and recommendations**	Electronic or digital resources recommended (6) Routine discussions around nutrition are supported (6)	“Questionnaires promote self-questioning behaviour rather than “being told” by a midwife”. (Midwife, 108)“I feel nutrition is so important and we should all be focusing on it more…Also as GPs we are not always well informed as not taught much about nutrition” (GP, 120).

*n* = number of coded comments.

## Data Availability

Anonymized data available on request due to restrictions. The data are not publicly available due ethical and privacy reasons as postcode and identifiable data were also collected in the survey.

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
