# Peer review of "Acceptability of the FIGO Nutrition Checklist in Preconception and Early Pregnancy to Assess Nutritional Status and Prevent Excess Gestational Weight Gain: A Study of Women and Healthcare Practitioners in the UK"

_nutrients, 2022, doi:10.3390/nu14173623_

Round 1

Reviewer 1 Report

The paper was clearly presented, and the results are very important for clinical practice.

Discussion should be improved, with the explanation of the paper's novelty.

Reviewer 2 Report

This manuscript evaluated the acceptability of using a nutrition checklist (FIGO) for women and HCPs in the UK and routine discussion of nutrition and weight in clinical care. The results showed that Routine discussion of nutrition was considered important by both groups, however, results suggest the concept and content of the checklist were acceptable to women and HCPs, while nutrition is not regularly discussed in perinatal visits in the UK. This study is very interesting and useful for the practice of nutritional screening in clinics. But the description of methods and results is not clear and lack of logistic, the following comments need to be addressed clearly.

For the method part of the manuscript,

1) the process of the study is not clear, a flow chart is needed.

2)The study materials of questionnaires is not well described, the structure and main items of questionnaires should be given.

3)The process of data collection needs more detailed information, eg. where and how to use the mixture of convenience, purposive and snowball sampling.

4) There is no ethical statement. Is the study approved by ethical committee?

For the result part of the manuscript,

1) The whole results is not arranged logistically and is too long, which need to be restructured and concise deeply. For example, the order of the results can be shown firstly the participant characteristics, secondly the acceptability of the checklist, thirdly routine discussion of nutrition.

2) There are too many tales and figures, which need to be reduced, combined or moved to supplement materials. 

Reviewer 3 Report

Thank you for the opportunity to review the article entitled: Acceptability of the FIGO Nutrition Checklist in Preconception and Early Pregnancy for Assessing Nutritional Status and Preventing Excessive Weight Gain During Gestational Weight Gain: A Study of Women and health professionals in the UK.

The authors Jacob and collaborators presented a study whose objective is: to evaluate the acceptability of 1) using a nutritional checklist designed by the International Federation of Gynecology and Obstetrics (FIGO) for the nutritional screening of women in the preconception period and early pregnancy and 2) routine discussion of nutrition and weight in clinical care.

This work has some limitations that are listed below:

1. As the results of the article are presented, they do not allow the acceptability of the checklist to be evaluated. A statistical test related to the objective of the work is not presented. Only descriptive results are presented that do not allow the evaluation of the acceptability of the survey according to what is stated in the objective.

2. The results do not support the conclusions.

3. In regression analyses, as the authors mention, the confidence intervals are too wide. In addition, the sample size is limited to support the associations (OR) presented. Therefore, the regression results presented may not be entirely valid.

4. The sample size is small and limits the validity of the results.

Round 2

Reviewer 2 Report

This manuscript has great improvement. One minor advice is listed below.

1. In the flow chart, please give the number of target groups, and clarify the name of target groups, eg.women, HCPS.

Reviewer 3 Report

Thanks again for the possibility of reviewing the article entitled: Acceptability of the FIGO Nutrition checklist in preconception and early pregnancy to assess nutritional status and prevent excess gestational weight gain: a study of women and healthcare practitioners in the UK

The authors were kind enough to respond to each of my comments, in addition to acknowledging and supporting the limitations of the study. I consider that the article has improved compared to its latest version, so it should be accepted.